# Factors Related to Psychological Well-Being as Moderated by Occupational Class in Korean Self-Employed Workers

**DOI:** 10.3390/ijerph19010141

**Published:** 2021-12-23

**Authors:** Jungsun Park, Hanjun Kim, Yangho Kim

**Affiliations:** 1Department of Occupational Health, Catholic University of Daegu, Gyeongsan 38430, Korea; jsunpark@chol.com; 2Department of Occupational and Environmental Medicine, Ulsan University Hospital, University of Ulsan College of Medicine, Ulsan 44033, Korea; 0734979@uuh.ulsan.kr

**Keywords:** work stressor, occupational class, well-being, self-employed, moderation

## Abstract

We examined factors related to the psychological well-being of self-employed workers in Korea, and the moderation of these effects by occupational class. This secondary analysis examined the data of 14,454 self-employed individuals from the fifth Korean Working Conditions Survey (2017). In all occupational classes, psychological well-being score was greater in women, and increased with monthly income and the frequency of working at very high speed; there were lower mean scores in those who became self-employed out of necessity rather than personal choice; in addition, the score decreased as the number of musculoskeletal symptoms increased. The relationship of work factors with the psychological well-being of self-employed individuals also differed according to occupational class. In conclusion, our analysis indicated that self-employed workers in different occupational classes respond differently to identical stressors.

## 1. Introduction

During 2009, workers who were self-employed (independent workers who had no waged employees) accounted for about 10% of all workers in Europe, with the lowest percentage in Estonia (4%) and the highest percentage in Greece (21%). Analysis of the entire European Union (EU) indicated a slight decrease in the percentage of workers who were self-employed since 2005 [1]. Self-employed workers account for a greater proportion of all workers in Korea than in the EU, although Korea has also experienced a decrease in the proportion of such workers, from 24.1% in 1988 to 15.3% in 2016 [2]. The decrease in the proportion of self-employed workers in Korea may be due to a higher rate of business failure stemming from the ease of starting a self-employed business, which has led to heightened competition [3,4]. The decrease may also be partly due to a decrease in the number of farmers, most of whom are self-employed [5]. Thus, the decrease suggests a precariousness among the self-employed population.

An individual may prefer self-employment for several reasons. In particular, self-employment provides more autonomy and the ability to choose working hours, work location, and type of work; some workers also choose self-employment because it gives them the ability to transform a hobby into a job [6]. However, some people with regular employment are forced to accept self-employment following job loss [6].

The European Foundation for the Improvement of Living and Working Conditions (Eurofound) classified self-employed people in the EU as “employers” (23%), “small traders and farmers” (25%), “stable own-account workers” (26%), “vulnerable workers” (17%), or “concealed workers” (8%) [1]. They also showed that “stable own-account workers” had more education and greater income, and were more likely to be self-employed because of personal preference, in contrast to “small traders and farmers” [1]. The rapid expansion of the “gig economy” has seen increases in the numbers of freelancers with precarious employment as well as professional freelancers with higher levels of education and higher incomes [7].

Korean studies classify “vulnerable workers” and “concealed workers” as employees, and do not consider “employers” as self-employed. Thus, self-employed small shop/restaurant owners and independent farmers account for about 80% of self-employed workers in Korea [8]. These individuals are more likely to be self-employed because they lack alternatives, rather than out of personal preference.

We explored the demographic, socioeconomic, and work-related factors related to mental health in the self-employed. Demographic factors, such as age and gender, may affect mental health. Older people are at greater risk of developing mental health conditions because of the cumulative effect of numerous risk factors, including chronic illness and isolation [9]. Previous studies found that women had more adverse psychological symptoms than men; refs. [10,11,12] although, Schutte et al. [13] found that the effect of gender on mental health varied among countries in Europe. Socioeconomic status affects mental health. A previous study found that socioeconomic status had both a direct impact on rates of mental illness as well as an indirect impact through higher levels of economic hardship in low and middle income groups [14]. Recent studies in the self-employed found a positive association between health and financial performance usually measured in terms of earnings [15,16]. Several job quality indices influence mental health [17]. Among those indices, we focused on work-related factors, such as job type, working hours/working days/work pace, the reason for being self-employed, and work–life balance, when we considered greater work autonomy of the self-employed. We also dealt with work-related factors, such as interaction with angry clients and physical illness related to mental health, which differently affect the mental health of the self-employed depending on their occupational class.

A European study of the mental well-being of the self-employed compared employers with other self-employed individuals (farmers, dependent freelancers, and own-account workers). Their results indicated that farmers, dependent freelancers, and own-account workers had worse mental well-being than medium-to-big employers [18]. A study of farmers in the UK reported that males, those who were 45 to 64 years old, the self-employed or those not in paid employment, those with non-supervisory positions, and those living in rural areas had higher mean scores on the General Health Questionnaire 12 (GHQ-12) than corresponding subgroups from the non-farming population [19]. Our previous study showed that, among self-employed individuals, men were more likely to engage in manual labor, and women were more likely to have service and sales jobs. We also found that self-employed individuals who performed office work had more education, higher incomes, and the least exposure to physical, chemical, and ergonomic workplace hazards. In contrast, manual workers were older, less educated, had lower incomes, had greater exposure to workplace hazards, and had more health problems [20]. Thus, comparing workers in different occupational classes may be important for understanding the mental well-being of the self-employed population.

There is limited knowledge of the factors that affect the mental health of the self-employed population. Stephan suggested that the effect of work on the mental health and well-being of entrepreneurs should be examined using different models because these individuals have high autonomy; they often have the options of deciding what work to perform, when to perform work, and whom to work with, and they often have demanding work that requires great effort and concentration [21]. Bradley and Roberts found that high work demands and long work hours led to greater satisfaction in starting entrepreneurs [22] because these individuals believed that hard work was an indication that the business was successful. In agreement, a large multi-national study reported a positive association of long working hours and improved mental health and well-being in entrepreneurs [23]. This suggests that entrepreneurs may see hard work and long working hours as a “challenge stressor” that provides them with opportunities for achievement [24]. Thus, the negative effect of highly demanding work decreases when there are sufficient resources, as described by the job demands–resources model. Importantly, these previous studies examined employers (self-employed individuals with employees) and self-employed individuals without employees.

Nikolova [25] reported that “necessity entrepreneurs” (individuals transitioning from unemployment to self-employment) had improved mental health but not physical health, but that “opportunity entrepreneurs” (individuals transitioning from regular employment to self-employment) had improved physical and mental health. These findings suggest that the reason for being self-employed may be a factor related to well-being in self-employed workers.

Self-employed individuals usually choose their work arrangements, thereby optimizing their autonomy. Thus, self-employment is more conducive to balancing work and family demands, especially for women [26,27]. A previous study on self-employed individuals including employers found that those with a good work–life balance were more likely to have higher job satisfaction and lower stress [26].

The issues affecting self-employed workers depend on their job characteristics, with small shop/restaurant owners, often dealing with angry customers, and farmers being more likely to have musculoskeletal symptoms. Previous studies showed that employed workers who always interacted with angry clients were more likely to experience depressive symptoms [28,29]; although, to our knowledge, no such study has been performed on the self-employed. A previous study on farmers showed that those with musculoskeletal symptoms were more likely to experience mental illness [30].

In the present study, we therefore used representative data from the Korea Working Conditions Survey (KWCS) to investigate the differences in psychological well-being between self-employed workers in different occupational classes. In addition, we investigated whether the influence of certain demographic, socioeconomic, and work-related factors differs depending on occupational class.

## 2. Methods

### 2.1. Data Source

This study was a secondary analysis of data collected during the fifth KWCS (June to September 2017), which was recently conducted by the Korea Occupational Safety and Health Agency (KOSHA) [31]. This triennial survey, which has high content validity and reliability [32], assesses working conditions, exposure to hazards, and work-related health problems at workplaces in Korea. The study population was representative of all Korean individuals who were at least 15 years old and worked during the survey period. A worker was defined as a person who worked for pay or profit for at least 1 h during the week before the interview. Retirees, the unemployed, homemakers, and students were excluded. This study was multistage and stratified, used a random sample design, and examined workers who lived in the same enumeration districts as those in the 2010 population and housing census in Korea. There were 50,205 in-person household interviews, and all data were weighted relative to the economically active population of Korea.

### 2.2. Study Subjects

After excluding foreign workers, employers, and waged workers, there were 14,454 Korean self-employed workers in the sample. A “self-employed” worker was defined as one who worked alone or with family members and had no employees. Family members who worked with a self-employed individual were not included as study subjects. The data from the KWCS are registered as government-approved statistics, and are strictly managed by the government, which guarantees confidentiality and anonymity.

### 2.3. Measurements

Most of the variables used in the present study (KWCS) originated from those in European Working Conditions Surveys (EWCS) [32].

#### 2.3.1. Dependent Variable

Self-reported well-being was determined using a survey developed by the World Health Organization—the WHO-5 Well-Being Index [33]. This index has five questions that ask participants to indicate their well-being during the two previous weeks. In particular, they were asked if they (a) felt cheerful and in good spirits, (b) felt calm and relaxed, (c) felt active and vigorous, (d) woke up feeling fresh and rested, and (e) had a daily life that was filled with things that interested them. Each item had a score of 0 to 5 points, the maximum score was 25, and a higher score indicated better psychological well-being. A total score of 13 or less indicated poor psychological well-being and the possibility of depression, and a score above 13 indicated good psychological well-being. Topp et al. cited several studies that use a cut-off of 50% (equivalent to a raw score of 13) as a “screening diagnosis” for depression when examining psychosocial problems related to the work environment [34].

#### 2.3.2. Independent Variables

Data on demographic, socioeconomic, and work-related psychosocial factors were recoded. The demographic factors included in the analysis were sex and age in years (<40, 40–49, 50–59, 60 or more). The socioeconomic factors included in the analysis were education (no high school graduation, high school graduation, college or more), monthly income in USD (<1000, 1000–2000, 2000–3000, 3000–4000, 4000 or more), and occupational class.

The occupational classes were “professionals and related workers”, “service and sales workers”, “agriculture, forestry and fisheries workers”, or “other manual workers”. After categorization of work as manual or non-manual, manual workers were further classified as “skilled workers related to agriculture, forestry and fisheries,” or “other manual workers”. The “other manual workers” were classified into 3 of the 9 major groups in the KSCO [35] and the ISCO [36] as “craft and related trade workers”, “workers related to equipment, machine operating and assembling”, and “elementary occupations” (unskilled manual workers). Non-manual workers were categorized as “managers, professionals and related workers, and clerks” or “service and sales workers.” “Service and sales work” often requires emotional labor, and the following definition of this term was adopted from a previous study: “the management of feelings to create a publicly observable facial and bodily display” [37]. “Service and sales work” is therefore different from other types of non-manual work that requires cognitive skills, such as that performed by “managers, professionals and related workers” and “clerks”.

Initial analysis of our 14,454 self-employed study subjects indicated that 26.2% were “sales workers”, 25.9% were “agriculture, forestry and fisheries workers”, 18.4% were “service workers”, 9.9% were “craft and related trade workers”, and the others were in other fields (Table 1). Only 0.1% of the study subjects were “managers” and 0.5% were “clerks”, so we excluded these groups from analysis. Thus, we classified the study subjects into 4 occupational classes: “professionals and related workers” (hereinafter—“professionals”), “service and sales workers” (hereinafter—“shop/restaurant owners”), “skilled workers related to agriculture, forestry and fisheries,” (hereinafter—“farmers”—because farmers accounted for 95.7% of this class), and “other manual workers” (hereinafter—“craftsmen”).

The present study used the following variables to measure high quantitative demands: long working hours, long working days, and working at very high speed. Working time (long working hours and long working days) is often included within the variable “working time quality”, whereas “working at very high speed” is often part of a scale called “work intensity”, which combines the pace of work and work pressure [17].

However, all these variables, when combined with greater work autonomy, lead to high work performance, and then high financial performance, among self-employed people. Thus, these variables were used together in the present study. Average weekly working time was classified as short working hours (SWH, <40 h), moderate working hours (MWH, 40–47 h), long working hours (LWH, 48–59 h), or excessively long working hours (ELWH, ≥60 h). Subjects were also classified as working fewer than 6 days per week or 6 or more days per week. The frequency of working at very high speed was assessed by asking: “Does your job involve working at very high speed?” The response options were “always”, “almost always”, “75% of the time”, “50% of the time”, “25% of the time”, “almost never”, or “never”. For analysis, the first 2 responses were classified as “always”; the next 3 responses as “frequently”; and the last 2 responses as “never”.

Work life balance, a job quality index related to the autonomy of self-employed workers, was assessed by asking: “In general, how do your working hours fit in with your family or social commitments outside work?” The response options were: “very well”, “well”, “not very well”, or “not at all well”. For analysis, the first 2 responses were classified as “good” and the last 2 responses as “bad”. The motivation for being self-employed was classified as “personal choice” or “no other alternative”.

Among the variables used to measure the emotional demands of jobs, workers who interacted with angry clients were identified by asking “Does your job involve interacting with angry customers, patients or students?” The response options were “always”, “almost always”, “75% of the time”, “50% of the time”, “25% of the time”, “almost never”, or “never”. For analysis, the first 2 responses were classified as “always”; the second 3 responses as “frequently”; and the last 2 responses as “never”.

We used musculoskeletal symptoms as a risk factor for depression. Workers with musculoskeletal symptoms were identified by asking “Have you suffered from back pain, upper limb pain, or lower limb pain during the past 12 months?” The response options were “yes” or “no”. Each musculoskeletal symptom was separately recorded as “back pain”, “upper extremity pain”, or “lower extremity pain”, and the number of musculoskeletal symptoms ranged from 0 to 3.

### 2.4. Statistical Analysis

For the analysis of study participants in different occupational classes, the chi-square test was used to compare categorical variables (demographic, socioeconomic, and psychosocial work factors) and the analysis of variance (ANOVA) test was used to compare continuous variables (age and psychological well-being score). Multiple linear regression was used to determine adjusted regression coefficients (B) and their 95% CIs for the relationship of psychological well-being with the following variables: demographic (age and sex) and socioeconomic factors (education, income, and occupational class), work-related factors (weekly working hours, weekly working days, and motivation for becoming self-employed, work–life balance, working at very high speed, musculoskeletal symptoms, and interacting with angry clients). These data were also analyzed using a moderated regression model with the Hayes PROCESS macro for SPSS (SPSS Inc., Chicago, IL, USA) [38,39]. Simple moderation analysis was used to determine whether the effect of various independent variables on poor psychological well-being varied in magnitude and nature as a function of occupational class (moderator). This analysis identified statistical interactions between independent variables that were predictors and those that were moderators (occupational class), and the strength and direction of these effects on psychological well-being after adjustment for other independent variables. List-wise deletion was used to handle missing data; a case was dropped from an analysis if it had a missing value for at least one of the specified variables. A *p*-value below 0.05 was considered significant.

## 3. Results

We examined the characteristics of workers in different occupational classes (Table 2). Higher percentages of farmers were older and had less education, lower monthly incomes, more musculoskeletal symptoms, a good work–life balance, and poor psychological well-being (all *p* < 0.001). Farmers also had a greater mean age and a lower mean psychological well-being score that was below the cut-off score of 13 for poor psychological well-being. Higher percentages of shop/restaurant owners were women, had ELWH and more weekly working days, and interacted with angry clients. Higher percentages of craftsmen were men and worked at very high speed. Higher percentages of professionals became self-employed by personal choice.

Next, we performed multivariate linear analysis of factors associated with psychological well-being with adjustment for confounding (Table 3). The results indicated that farmers had lower psychological well-being than professionals. Psychological well-being was negatively associated with age and decreased as the number of musculoskeletal symptoms increased. However, psychological well-being increased with educational level and monthly income. We observed higher psychological well-being scores in individuals with a higher frequency of working at very high speed and working 6 or more days per week. By contrast, lower psychological well-being scores were observed in men, individuals who became self-employed because there was no alternative, individuals with a bad work–life balance, and individuals who always interacted with angry clients. The number of weekly working hours was unrelated to psychological well-being.

We performed a moderation analysis to determine whether the effects of various independent variables on poor psychological well-being varied in magnitude and nature as a function of occupational class (moderator) after adjustment for other independent variables. The results indicated that the effects of age, education level, working hours per week, working days per week, number of musculoskeletal symptoms, work–life balance, and interaction with angry clients on the psychological well-being score was affected by occupational class (*p*-interaction < 0.05). However, the effects of gender, monthly income, frequency of working at very high speed, and motivation for being self-employed on psychological well-being score was not affected by occupational class (*p*-interaction > 0.05).

We also examined the conditional effects of the different independent variables on the psychological well-being score after adjustment for other independent variables (Table 4; Figure 1a–g). Our analysis indicated that psychological well-being score was negatively associated with age in occupational classes other than craftsmen. Psychological well-being score was greater in shop/restaurant owners and manual workers (especially farmers) who had more education, but psychological well-being score was unrelated to education level in professionals. Psychological well-being score was greater in farmers with ELWH but was unrelated to weekly working hours in other classes. Psychological well-being score was greater in manual workers (especially farmers) who worked 6 or more days per week but was unrelated to weekly working days in non-manual workers. For all classes, psychological well-being score decreased as number of musculoskeletal symptoms increased, and this tendency was greater in professionals. Psychological well-being score was also greater in workers who had good work–life balance in all the occupational classes, especially shop/restaurant owners. Psychological well-being score was lower in professionals, shop/restaurant owners, and craftsmen who always interacted with angry clients. Because most farmers (93.5%) never interacted with angry clients, the statistical estimate for this variable may not be accurate.

In all occupational classes, psychological well-being score was greater in women, those with higher monthly incomes, those who always or frequently worked at very high speed, and those who became self-employed by personal choice.

## 4. Discussion

The present study of self-employed workers in Korea examined the effects of demographic, socioeconomic, and psychosocial work factors on the mental health of such workers.

The average age of self-employed individuals in Korea ranged from 49.4 to 69.4 years old, depending on the occupational class. Farmers and shop/restaurant owners account for 70% of the self-employed in Korea. Young people from Korea’s rural areas often move to cities to seek work, leaving only the elderly. Thus, the average age of farmers in Korea is very high (69.4 years old). Farmers in Korea are older than those in the EU, although the agricultural labor force is also ageing in EU. For example, only 11% of farm managers in the EU are under age 40, and almost one-third of them are 65 years old or more [40]. In addition, many self-employed workers who run small restaurants and shops in Korea retired from their primary jobs when they were in their fifties, and chose to run small restaurants and shops that do not require special skills. These factors contribute to the high average age of the self-employed in Korea. As described in the Introduction of this study, the number of self-employed people in Korea is gradually decreasing because the elderly people in rural areas die or retire from work, and an increasing number of small shop/restaurant owners fail in their businesses due to fierce competition.

### 4.1. Factors Related to Psychological Well-Being in the Self-Employed

We found that psychological well-being score in the self-employed increased as education level and monthly income increased. These findings are consistent with previous studies [10,11,13,41], which found dose–response relationships of education and income with mental health. However, we also found that psychological well-being score was greater in self-employed women than men. This is in contrast with several previous studies, which reported that women had more adverse psychological symptoms than men [10,11,12]. Notably, Schutte et al. [13] found that the effect of gender on mental health varied among countries in Europe. Korean society has a tradition of patriarchy, in that women tend to be less respected than men at home and in society. Because of this tradition, female self-employed individuals who own businesses in Korea tend to be those who have higher self-esteem and well-being than those who are temporary or part-time employees, or unemployed [42].

In agreement with previous studies, we found that self-employed individuals in all classes had greater psychological well-being scores if they had a good work–life balance and were self-employed by personal choice [26,43,44,45]. Additionally, in agreement with previous reports, psychological well-being score in the self-employed was lower in those who always interacted with angry clients [28,29,46].

Our analysis of the self-employed found that psychological well-being score increased as the frequency of working at very high speed increased, and that this score was greater in farmers who worked more than 60 h per week and in manual workers who worked 6 or more days per week. These unexpected findings may be because the self-employed typically have greater work autonomy, such as the freedom to choose their working hours, and because self-employed individuals may choose to work more hours to earn more money. High monthly income had a positive effect on psychological well-being in the self-employed. These economic rewards and feelings of achievement may increase the well-being of the self-employed despite their longer working hours, according to the “effort–reward imbalance model” [47]. Recent studies found a positive association between health and financial performance in the self-employed [15], usually measured in terms of earnings [16]. The self-employed typically have longer working hours than wage workers [8,48], but this does not have to lead to higher stress levels for the self-employed [49]. Working longer hours may indicate the business is doing well and has, for example, been associated with higher levels of work satisfaction among the self-employed [23]. These findings point to the need for further studies of the exact relationship between mental health and work performance in the self-employed.

### 4.2. Relationship of Occupational Class with Psychological Well-Being

The present study also identified differences among self-employed individuals in different occupational classes. Self-employed professionals had lower psychological well-being scores than those in other occupational classes when they always interacted with angry clients. It is possible that professionals are more sensitive to interactions with angry clients because they take great pride in their own accomplishments and professionalism, and do not like being challenged by angry clients [50,51].

Economic rewards from long working hours may increase psychological well-being in self-employed shop/restaurant owners. In addition, even if a female self-employed individual works alone in a small shop/restaurant, she will have high self-esteem because she owns the business. This high self-esteem may also increase psychological well-being [42].

Compared with self-employed workers in other occupational classes, farmers who live in rural areas lack easy access to the cultural and educational facilities and medical benefits provided in cities, which may reduce their self-esteem and hence their mental well-being. Farmers tend to be economically independent and work autonomously [1], and are motivated to increase their incomes by simply working more hours [7]. These rewards and job autonomy may be responsible for the high psychological well-being of farmers who work more. Farmers were more likely to have musculoskeletal symptoms, although this had less of an effect on their well-being scores than those in other occupational class, probably because farmers accept physically demanding labor as part of their work.

We found that higher percentages of craftsmen were male and always worked at very high speed. This group also had the second highest average monthly income (after professionals), and the second highest frequency of musculoskeletal symptoms and low well-being scores (after farmers). Although self-employed craftsmen are also manual workers (alike to farmers), they differ from farmers because they have higher incomes due to their special skills.

Self-employed workers in different occupational classes responded differently to identical work stressors. The present study is the first to demonstrate that the effects of demographic, socioeconomic, and work-related factors on psychological well-being in the self-employed population depend on occupational class. This novel finding may be due to differences in the individual characteristics of people in these different groups or due to the inherent characteristics of the different types of work [52].

Our finding that factors related to psychological well-being in the self-employed differed among occupational classes may have implications for employment and public health policies in Korea. Self-employed individuals have greater work autonomy, such as the freedom to choose their own working hours and methods of working, than waged employees. In addition, the self-employed can often work as much they want, assuming work is available. Hence, they may be more likely to increase their incomes by simply working more hours. The OSH regulations in Korea and many other countries were designed to protect waged workers, not the self-employed, and self-employed individuals may thus be more vulnerable to OSH problems. Therefore, we suggest that vulnerable self-employed individuals who are currently unprotected by OSH regulations should be eligible to receive mental health care from community healthcare services, such as regional mental health welfare centers. Community healthcare services may also be able to help them to improve the well-being of the self-employed. Specific measures based on occupational differences are also recommended. For example, farmers were more likely to have higher well-being scores when they worked more than 60 h per week, 6 or more days per week, and frequently work at high speed, indicating that they are motivated to increase their incomes by simply working more hours. However, they were also more likely to be old and have musculoskeletal symptoms, and are not protected by the occupational health system. Hence, preventative measures against musculoskeletal disorders targeting self-employed farmers should be implemented. For small-scale shop/restaurant owners, who were more likely to work extremely long hours, work 6 days or more per week, and have a poor work–life balance, consumers’ consensus on ensuring adequate rest and sleep for the self-employed should be attained across society.

The present study has several strengths. First, the study was based on representative survey data from the KWCS, which covers all adult workers in Korea and employs rigorous quality control measures. Second, the results obtained from the Korean population (KWCS) differed from those obtained in a European context (EWCS). For example, professional “stable own-account workers” were dominant in the EWCS, whereas precarious “small shop/restaurant owners and farmers” were dominant in the KWCS. The results of the present study were based on small shop/restaurant owners and farmers, and may serve as a suggestion for further studies in the European context.

This study has several limitations. First, our study had a cross-sectional design, so we cannot infer the causality of associations because unknown intermediary factors may be responsible. A study with a prospective design is needed to establish the causal bases for the associations identified here. Second, we partly relied on self-reported data rather than objective data. Our results should therefore be interpreted with caution, because self-reported questionnaires may have questionable validity.

## 5. Conclusions

Our study has two major conclusions. First, our analysis of factors related to the psychological well-being of self-employed workers indicated that psychological well-being score was greater in women and increased with monthly income and with the frequency of working at a very high speed. However, this well-being score was lower in those who became self-employed out of necessity rather than personal choice. Second, our analysis of the relationship of work factors with psychological well-being in the self-employed indicated that individuals in different occupational classes are unique, in that they respond differently to identical stressors. Thus, specific interventions based on occupational differences may be needed to improve the psychological well-being of self-employed workers.

## Figures and Tables

**Figure 1 ijerph-19-00141-f001:**
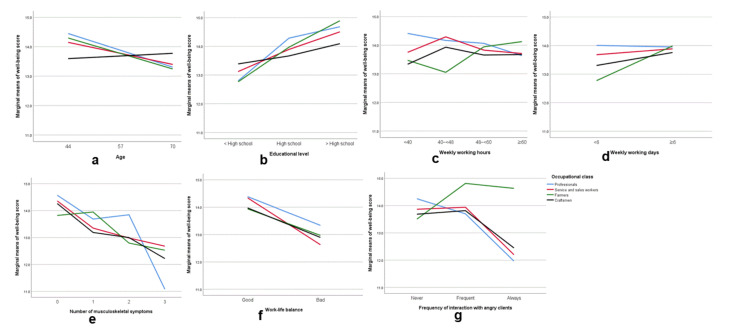
(**a**–**g**). Interaction between independent variables and occupational class in their influence on psychological well-being: (**a**) age; (**b**) educational level; (**c**) weekly working hours; (**d**) weekly working days; (**e**) number of musculoskeletal symptoms; (**f**) work–life balance; (**g**) frequency of interaction with angry clients.

**Table 1 ijerph-19-00141-t001:** Job categories of self-employed study subjects.

Category	*n*	(%)
Managers *	15	0.1
Professionals and related workers	1086	7.5
Clerks *	79	0.5
Service workers	2664	18.4
Sales workers	3780	26.2
Skilled workers related to agriculture, forestry, and fisheries	3742	25.9
Craft and related trade workers	1431	9.9
Workers related to equipment, machine operating, and assembling	1347	9.3
Elementary occupations	310	2.1
Total	14,454	100.0

* Excluded from subsequent analyses.

**Table 2 ijerph-19-00141-t002:** Demographic, socioeconomic, and psychosocial work-related factors of self-employed workers in different occupational classes.

		Professional	Shop/Restaurant Owner	Farmer	Craftsmen	Total	*p*-Value
Age, years	<40	200 18.4%	836 13.0%	35 0.9%	198 6.4%	1269 8.8%	<0.001
	40–49	345 31.8%	1567 24.3%	167 4.5%	567 18.4%	2646 18.4%	
	50–59	360 33.1%	2371 36.8%	511 13.7%	1196 38.7%	4438 30.9%	
	>60	181 16.7%	1670 25.9%	3029 80.9%	1127 36.5%	6007 41.8%	
Age	Years	49.4 ± 10.5	52.5 ± 11.0	69.4 ± 10.8	56.0 ± 10.3	57.4 ± 13.0	<0.001
Gender	Men	539 49.6%	2030 31.5%	2074 55.4%	2305 74.6%	6948 48.4%	<0.001
	Women	547 50.4%	4414 68.5%	1668 44.6%	783 25.4%	7412 51.6%	
Education	<High school	26 2.4%	1161 18.1%	2664 71.3%	791 25.6%	4642 32.4%	<0.001
	High school	283 26.1%	3524 54.8%	892 23.9%	1784 57.8%	6483 45.2%	
	>High school	775 71.5%	1746 27.1%	179 4.8%	512 16.6%	3212 22.4%	
Monthly income, USD	<1000	35 3.3%	265 4.2%	1599 43.2%	207 6.8%	2106 14.9%	<0.001
	1000 to <2000	181 16.9%	1513 23.9%	1154 31.2%	664 21.7%	3512 24.8%	
	2000 to <3000	294 27.5%	2221 35.1%	600 16.2%	934 30.6%	4049 28.6%	
	3000 to <4000	327 30.6%	1515 23.9%	213 5.8%	740 24.2%	2795 19.7%	
	≥4000	231 21.6%	818 12.9%	133 3.6%	511 16.7%	1693 12.0%	
Weekly working hours	<40	241 22.3%	489 7.6%	1877 50.5%	429 13.9%	3036 21.2%	<0.001
	40 to <48	226 20.9%	591 9.2%	769 20.7%	510 16.6%	2096 14.7%	
	48 to <60	352 32.5%	1964 30.6%	640 17.2%	968 31.5%	3924 27.4%	
	≥60	264 24.4%	3380 52.6%	431 11.6%	1170 38.0%	5245 36.7%	
Weekly working days	<6	389 35.9%	695 10.8%	1486 40.0%	855 27.8%	3425 24.0%	<0.001
	≥6	695 64.1%	5720 89.2%	2229 60.0%	2222 72.2%	10,866 76.0%	
Working at a very high speed	Always	78 7.2%	671 10.4%	353 9.4%	521 16.9%	1623 11.3%	<0.001
	Frequent	316 29.1%	2181 33.9%	1027 27.4%	1282 41.5%	4806 33.5%	
	Never	692 63.7%	3591 55.7%	2362 63.1%	1285 41.6%	7930 55.2%	
Musculoskeletal symptoms (*n*)	0	893 82.2%	4149 64.5%	1246 33.3%	1615 52.4%	7903 55.1%	<0.001
	1	127 11.7%	876 13.6%	570 15.2%	572 18.5%	2145 15.0%	
	2	48 4.4%	950 14.8%	915 24.5%	583 18.9%	2496 17.4%	
	3	18 1.7%	462 7.2%	1007 26.9%	315 10.2%	1802 12.6%	
Motivation	Personal choice	977 96.0%	5502 93.2%	2846 90.6%	2591 92.7%	11,916 92.7%	<0.001
	No other alternative	41 4.0%	401 6.8%	297 9.4%	204 7.3%	943 7.3%	
Work–life balance	Good	870 80.3%	3784 58.8%	3144 84.2%	2047 66.4%	9845 68.7%	<0.001
	Bad	213 19.7%	2648 41.2%	591 15.8%	1037 33.6%	4489 31.3%	
Dealing with angry clients	Always	59 5.4%	374 5.8%	44 1.2%	129 4.2%	606 4.2%	<0.001
	Frequent	214 19.7%	1599 24.8%	198 5.3%	711 23.0%	2722 19.0%	
	Never	811 74.8%	4467 69.4%	3494 93.5%	2247 72.8%	11,019 76.8%	
Poor psychological well-being	No	824 75.9%	4420 68.7%	1870 50.1%	1987 64.4%	9101 63.5%	<0.001
	Yes	262 24.1%	2013 31.3%	1859 49.9%	1098 35.6%	5232 36.5%	
Psychological well-being score		15.2 ± 5.0	14.2 ± 5.1	11.8 ± 5.7	13.7 ± 5.2	13.6 ± 5.4	<0.001

—Here, and below: classified as no completion of high school, completion of high school, or post-high school education.

**Table 3 ijerph-19-00141-t003:** Multivariate linear regression analysis of the association of psychological well-being score with occupational class, demographic factors, socioeconomic factors, and psychosocial work-related factors in the self-employed workers (*n* = 12,573).

Variables	Classification	B Coefficient (95%CI)
Occupational class	Professional	0 (reference)
	Shop/restaurant owner	−0.235 (−0.599, 0.129)
	Farmer	−0.437 (−0.868, −0.005) *
	Craftsmen	−0.377 (−0.774, 0.021)
Age, years		−0.026 (−0.037, −0.016) ***
Gender	Men	0 (reference)
	Women	0.387 (0.188, 0.586) ***
Education	<High school	0 (reference)
	High school	0.797 (0.529, 1.064) ***
	>High school	1.365 (1.008, 1.721) ***
Monthly income, USD	<1000	0 (reference)
	1000 to <2000	1.269 (0.934, 1.603) ***
	2000 to <3000	1.613 (1.249, 1.977) ***
	3000 to <4000	1.608 (1.208, 2.008) ***
	≥4000	2.258 (1.824, 2.692) ***
Weekly working hours	<40	0 (reference)
	40 to <48	0.116 (−0.199, 0.432)
	48 to <60	0.171 (−0.147, 0.488)
	≥60	0.074 (−0.257, 0.405)
Weekly working days	Fewer than 6	0 (reference)
	6 or more	0.650 (0.392, 0.909) ***
Working at very high speed	Never	0 (reference)
	Frequent	0.774 (0.575, 0.973) ***
	Always	1.893 (1.593, 2.194) ***
No. of musculoskeletal symptoms	0	0 (reference)
	1	−0.755 (−1.018, −0.493) ***
	2	−1.283 (−1.540, −1.026) ***
	3	−1.697 (−2.006, −1.389) ***
Motivation	Personal choice	
	No other alternative	−0.762 (−1.106, −0.417) ***
Work–life balance	Good	0 (reference)
	Bad	−1.405 (−1.612, −1.198) ***
Interaction with angry clients	Never	0 (reference)
	Frequently	0.134 (−0.103, 0.370)
	Always	−1.441 (−1.883, −0.999) ***

* *p* < 0.05 vs. reference; *** *p* < 0.001 vs. reference.

**Table 4 ijerph-19-00141-t004:** Conditional effect (B coefficients [95% CI]) of independent variables on psychological well-being for each occupational class after adjustment for age, sex, education, monthly income, working hours, working days, and other work-related psychosocial factors. (*n* = 12,573).

		Interaction (Independent Variable × Occupational Class)			
		Occupational Class			
Independent Variable		Professional	Shop/Restaurant Owner	Farmer	Craftsmen
Age	Years	−0.044 (−0.074, −0.013) **	−0.029 (−0.042, −0.015) ***	−0.040 (−0.059, −0.022) ***	0.007 (−0.013, 0.027)
Education	<High school	0 (reference)	0 (reference)	0 (reference)	0 (reference)
	High school	1.489 (−0.589, 3.567)	0.771 (0.387, 1.155) ***	1.204 (0.753, 1.656) ***	0.279 (−0.189, 0.747)
	>High school	1.885 (−0.140, 3.910)	1.383 (0.923, 1.842) ***	2.124 (1.273, 2.976) ***	0.705 (0.075, 1.334) *
Weekly working hours	<40	0 (reference)	0 (reference)	0 (reference)	0 (reference)
	40 to <48	−0.245 (−1.206, 0.715)	0.540 (−0.110, 1.190)	−0.420 (−0.897, 0.056)	0.595(−0.094, 1.284)
	48 to <60	−0.347 (−1.246, 0.551)	0.077 (−0.468, 0.621)	0.471 (−0.060, 1.003)	0.321(−0.306, 0.948)
	≥60	−0.777 (−1.740, 0.180)	−0.042 (−0.571, 0.488)	0.653 (0.040, 1.266) *	0.338 (−0.284, 0.960)
Weekly working days	<6	0 (reference)	0 (reference)	0 (reference)	0 (reference)
	≥6	−0.054 (−0.744, 0.637)	0.206 (−0.251, 0.663)	1.216 (0.833, 1.599) ***	0.454 (0.007, 0.901) *
Musculoskeletal symptoms (*n*)	0	0 (reference)	0 (reference)	0 (reference)	0 (reference)
	1	−0.886 (−1.879, 0.108)	−1.009 (−1.407,−0.612) ***	0.126 (−0.417, 0.668)	−1.073 (−1.583, −0.564) ***
	2	−0.726 (−2.310, 0.859)	−1.376 (−1.758, −0.993) ***	−1.028 (−1.507, −0.548) ***	−1.262 (−1.781, −0.743) ***
	3	−3.489 (−6.158, −0.820) *	−1.680 (−2.206, −1.153) ***	−1.293 (−1.777, −0.810) ***	−2.042 (−2.695, −1.389) ***
Work–life balance	Good	0 (reference)	0 (reference)	0 (reference)	0 (reference)
	Bad	−1.050 (−1.851, −0.249) *	−1.711 (−1.986, −1.437) ***	−0.968 (−1.475, −0.460) ***	−1.077 (−1.487, −0.666) ***
Interaction with angry clients	Never	0 (reference)	0 (reference)	0 (reference)	0 (reference)
	Frequently	−0.553 (−1.360, 0.255)	0.075 (−0.236, −0.386)	1.308 (0.500, 2.116) **	0.131 (−0.326, 0.587)
	Always	−2.286 (−3.658, −0.913) **	−1.666 (−2.225, −1.106) ***	1.126 (−0.501, 2.752)	−1.237 (−2.198, −0.277) *

* *p* < 0.05 vs. reference; ** *p* < 0.01 vs. reference; *** *p* < 0.001 vs. reference.

## Data Availability

Data available in a publicly accessible repository (https://oshri.kosha.or.kr/oshri/researchField/downWorkingEnvironmentSurvey.do, accessed on 16 November 2021).

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
