# Peer review of "Factors Related to Psychological Well-Being as Moderated by Occupational Class in Korean Self-Employed Workers"

_ijerph, 2021, doi:10.3390/ijerph19010141_

Round 1
Reviewer 1 Report
The topic is very much relevant nowadays. As per the research methods, the authors used secondary data. It need justification of choosing a particular time period as for this study it is June to September 2017. It also needs to consider it is now 2021 and this study is relevant if considering data from 2017. It can also see that statistical method is used in this study. This needs to justify whether Chi-square is used or t-test is used. Conclusion of the study can be categorised under different heading.
Author Response
Reviewer 1
Comment 1) The topic is very much relevant nowadays.
As per the research methods, the authors used secondary data. It need justification of choosing a particular time period as for this study it is June to September 2017.
Response) This study was a secondary analysis of data collected during the fifth KWCS (June to September 2017), which was conducted by the Korea Occupational Safety and Health Agency (KOSHA) every three years. This survey is the most recent one, which is open to the public.
We added ‘recently’ to the sentence.
This study was a secondary analysis of data collected during the fifth KWCS (June to September 2017), which was recently conducted by the Korea Occupational Safety and Health Agency (KOSHA)
Comment 2) It also needs to consider it is now 2021 and this study is relevant if considering data from 2017.
Response) The authors do not think that the unique characteristics of self-employed workers have changed significantly although the proportion of self-employed may have changed.
Comment 3) It can also see that statistical method is used in this study. This needs to justify whether Chi-square is used or t-test is used.
Response) We revised them as follows.
For analysis of study participants in different occupational classes, the chi-square test was used to compare categorical variables (demographic, socioeconomic, and psychosocial work factors) and the analysis of variance (ANOVA) test was used to compare continuous variables (age and psychological well-being score).
Comment 4) Conclusion of the study can be categorised under different heading.
Response) We categorized the conclusion as follows.
Our study has two major conclusions. First, our analysis of factors related to the psychological well-being of self-employed workers indicated that psychological well-being score was greater in women, and increased with monthly income and with the frequency of working at a very high speed. However, this well-being score was lower in those who became self-employed out of necessity rather than personal choice. Second, our analysis of the relationship of work factors with psychological well-being in the self-employed indicated that individuals.
Reviewer 2 Report
The present article investigates factors that correlate with psychological well-being among the self-employed in Korea. I found the article well-written and easy to understand. A clear strength of the article is the data, which is representative of the Korean population. The methods used are simple, which is a good thing, as articles of this type sometimes uses very sophisticated methods, although the data is not very good.
The authors state as a weakness that the data comes from Korea, and may not be directly applicable to other economies. However, this is not really a weakness but an opportunity. Korea is a rather large economy, and results are therefore even for that reason quite interesting. The authors should instead try to exploit the Korean aspect of the data more. For instance, the average age of farmers in the sample is very high. This must be a particularity of Korea that needs more explanation. Why is this the case? Is this a lot higher that in say, Western Europe? Please provide a comparison. Overall, the average age of the sample of self-employed is also very high. Please provide an explanation for this. Is self-employment something that will rapidly decline in the Korean labour market, as the age of the self-employed is so high? Yet another question concerns the interesting result regarding women, who were found to have higher psychological well-being than men. I judge that some more discussion regarding this result would be warranted in the article.
On the other hand, the discussion part of the article could be shortened. Now, quite a bit of space is devoted to discussion or repeating results that are not particularly different to what is found in other studies. Also, figures 1-7 are perhaps not that interesting. Could they perhaps be combined or event removed. The results are rather clear anyway.
Author Response
Reviewer 2
The present article investigates factors that correlate with psychological well-being among the self-employed in Korea. I found the article well-written and easy to understand. A clear strength of the article is the data, which is representative of the Korean population. The methods used are simple, which is a good thing, as articles of this type sometimes uses very sophisticated methods, although the data is not very good.
Comment 1)The authors state as a weakness that the data comes from Korea, and may not be directly applicable to other economies. However, this is not really a weakness but an opportunity. Korea is a rather large economy, and results are therefore even for that reason quite interesting.
Response) We deleted the sentence described in the limitation of the study.
Comment 2) The authors should instead try to exploit the Korean aspect of the data more. For instance, the average age of farmers in the sample is very high. This must be a particularity of Korea that needs more explanation. Why is this the case? Is this a lot higher that in say, Western Europe? Please provide a comparison. Overall, the average age of the sample of self-employed is also very high. Please provide an explanation for this.
Response) We provide an explanation for this ageing issue in farmers, and compared it with Europe.
The average age of self-employed individuals in Korea ranged from 49.4 to 69.4 years-old, depending on the occupational class. Farmers and shop/restaurant owners account for 70% of the self-employed in Korea. Young people from Korea's rural areas often move to cities to seek work, leaving only the elderly. Thus, the average age of farmers in Korea is very high (69.4 years-old). Farmers in Korea are older than those in the EU, although the agricultural labor force is also ageing in EU. For example, only 11% of farm managers in the EU are under age 40, and almost one-third of them are 65 years-old or more [40]. In addition, many self-employed workers who run small restaurants and shops in Korea retired from their primary jobs when they were in their fifties, and chose to run small restaurants and shops that do not require special skills. These factors contribute to the high average age of the self-employed in Korea.
Comment 3) Is self-employment something that will rapidly decline in the Korean labour market, as the age of the self-employed is so high?
Response) We addressed this issue.
As described in the Introduction, the number of self-employed people in Korea is gradually decreasing because the elderly people in rural areas die or retire from work, and an increasing number of small shop/restaurant owners fail in their businesses due to fierce competition.
Comment 4) Yet another question concerns the interesting result regarding women, who were found to have higher psychological well-being than men. I judge that some more discussion regarding this result would be warranted in the article.
Response) We added more discussion on this issue.
Korean society has a tradition of patriarchy, in that women tend to be less respected than men at home and in society. Because of this tradition, female self-employed individuals who own businesses in Korea tend to be those who have higher self-esteem and well-being than those who are temporary or part-time employees, or unemployed.
Comment 5) On the other hand, the discussion part of the article could be shortened. Now, quite a bit of space is devoted to discussion or repeating results that are not particularly different to what is found in other studies.
Response) We shortened Discussion section by deleting discussion or repeating results that are not particularly different to what is found in other studies.
Comment 6) Also, figures 1-7 are perhaps not that interesting. Could they perhaps be combined or event removed. The results are rather clear anyway.
Response) We combined the figures.